# Serum Visfatin/eNAMPT as a Biomarker in Pancreatic and Small Intestine Neuroendocrine Tumors: A Cross-Sectional Study and Future Perspectives

**DOI:** 10.3390/cancers17142343

**Published:** 2025-07-15

**Authors:** Paweł Komarnicki, Adam Maciejewski, Jan Musiałkiewicz, Michalina Czupińska, George Mastorakos, Marek Ruchała, Paweł Gut

**Affiliations:** 1Department of Endocrinology, Metabolism and Internal Diseases, Poznan University of Medical Sciences, Przybyszewskiego 49, 60-355 Poznań, Polandmruchala@ump.edu.pl (M.R.);; 2Unit of Endocrinology, Diabetes Mellitus and Metabolism, Aretaieion Hospital, Medical School, National and Kapodistrian University of Athens, 157 72 Athens, Greece

**Keywords:** biomarkers, neuroendocrine neoplasms, nicotinamide phosphoribosyltransferase, pre-B-cell colony-enhancing factor 1, visfatin

## Abstract

This study demonstrates for the first time the potential utility of serum visfatin/eNAMPT measurements as a diagnostic biomarker for neuroendocrine tumors (NETs). Visfatin/eNAMPT concentrations are increased in patients with metastatic pancreatic and small intestinal NETs and show potential in distinguishing NETs from controls, regardless of tumor and patient characteristics. As a biomarker, visfatin could act as a bridge between currently used imperfect monoanalytes such as chromogranin A and expensive multianalytes such as NETest. Although serum visfatin levels might not directly reflect tissue NAMPT expression, we also review visfatin’s role in a therapeutic setting in the context of NAMPT inhibitors, researched in previous studies.

## 1. Introduction

Neuroendocrine neoplasms (NENs) are highly heterogeneous tumors arising from endocrine and nervous system cells. NENs are divided into neuroendocrine tumors (NETs), which are usually well-differentiated, and poorly differentiated neuroendocrine carcinomas (NECs). NETs can be characterized based on their primary site (the gastrointestinal tract, pancreas, and lungs are the most common locations), proliferative activity (WHO grades depend on the tumor’s Ki-67 index and mitotic count), and hormonal secretion (functioning vs. non-functioning) [1]. NENs are generally rare, accounting for less than 2% of all malignancies. NET prevalence ranges between 2.5 and 8.35 cases per 10,000 and remains below 200,000 cases/year in the U.S., meeting the FDA’s rare disease criteria. However, NET incidence has been rising in recent years, partly due to the improvement in the accuracy and availability of medical imaging [2,3]. Despite significant advances in understanding NENs’ etiology, pathogenesis, and therapy, several issues remain unsolved. In particular, identifying reliable diagnostic and prognostic biomarkers for early disease detection and effective, tailored treatment remains an unmet need [4,5]. NET biomarkers can be divided into non-specific (chromogranin A, neuron-specific enolase, and beta-human chorionic gonadotropin) and tumor-specific analytes (insulin, gastrin, vasoactive intestinal peptide, glucagon, somatostatin, serotonin, and 5-hydroxyindoleacetic acid) [2,6]. Unfortunately, despite initial promise, no single biomarker has demonstrated optimal diagnostic and prognostic accuracy for a disease as diverse as NETs. Recent advances led to the development of novel biomarkers, utilizing liquid biopsy and multiomics-based approaches, such as NETest, microRNAs, cell-free DNA, and circulating tumor cells [2]. Despite improved performance, they are often expensive or not readily available. Thus, reliable biochemical markers for both diagnostic and prognostic purposes are still needed [2].

Visfatin, also known as nicotinamide phosphoribosyltransferase (NAMPT) or pre-B-cell colony-enhancing factor 1 (PBEF1), is a rate-limiting enzyme in the NAD+ (nicotinamide adenine dinucleotide) synthesis salvage pathway [7]. NAD+ is an essential cofactor involved in redox reactions and serves as a substrate for multiple signaling enzymes, including poly(ADP-ribose) polymerases, cyclic ADP-ribose synthetases, sirtuins, and CD38 [8]. The role of NAD+ in cancer is well-established, as cancer cells utilize it extensively for metabolism and survival [8]. NAMPT exists in two distinct forms, namely intracellular NAMPT (iNAMPT), primarily found in the cytosol and nucleus, and extracellular NAMPT (eNAMPT)—the term visfatin is preserved for this form. iNAMPT is the predominant form in physiological conditions, and eNAMPT/visfatin constitutes only about 1% of total NAMPT levels [9]. iNAMPT is responsible for the enzymatic regulation of NAD+ biosynthesis, whereas eNAMPT functions as a proinflammatory cytokine/adipocytokine. eNAMPT activates Toll-like receptor 4 (TLR4), stimulates B-lymphocytes, monocytes, and neutrophils, and increases the secretion of proinflammatory cytokines (e.g., IL-1β, IL-1Ra, IL-6, CXCL8, IL-10, TNF-α) and chemokines (CCL2, CCL3, CCL18, and CCL20) [7].

In the context of biomarker research, eNAMPT/visfatin has attracted increasing interest. Visfatin was previously investigated for its insulin-mimetic properties in metabolic disorders. It is predominantly secreted by adipocytes, macrophages, lymphocytes, and neoplastic cells [9]. Although increased circulating visfatin levels have been reported in several malignancies, its association with NETs remains unexplored [10,11]. 

We aimed to assess, for the first time, the utility of serum visfatin concentration as a diagnostic marker in NETs, specifically small intestine NETs (siNETs) and pancreatic NETs (panNETs). We seek to explore the involvement of visfatin/eNAMPT in NETs pathogenesis, potentially paving the way for personalized and targeted treatment strategies. Finally, we aimed to determine if circulating visfatin correlates with patient and tumor characteristics including WHO grade, NET primary site, sex, and age.

## 2. Materials and Methods

### 2.1. Study Design and Patient Population

This was a single-center, cross-sectional study, conducted at the Department of Endocrinology, Metabolism, and Internal Diseases at Poznan University of Medical Sciences in Poznan, Poland. Between February and November 2024, we recruited 77 patients with histologically confirmed pancreatic (33) or small intestinal (44) NETs and 29 controls. Patient demographics and tumor characteristics were collected at the time of inclusion. All patients with NETs were diagnosed with metastatic or locally advanced disease and were treated with long-acting somatostatin analogs (SSA) at the time of the sampling (lanreotide, Somatuline Autogel 120 mg once every 4 weeks; Ipsen, Paris, France). WHO grade was recorded based on a histopathological evaluation of tumor tissue at the time of the diagnosis. Exclusion criteria included prior or current systemic therapy for NETs (e.g., everolimus, sunitinib, chemotherapy, peptide receptor radionuclide therapy), active malignancy other than NETs, diabetes, renal or liver failure, untreated or unregulated hyperlipidemia, hypertension, thyroid dysfunction, recent surgery (up to 3 months before the recruitment), obesity (defined as BMI ≥ 30), autoimmune diseases requiring biological treatment, ECOG/WHO performance status ≥ 3, acute infections or other acute conditions at the time of the sampling, pregnancy, or lactation. Participants with missing data were excluded from the study. Controls without a NET diagnosis were recruited from among patients hospitalized at the same department. These were patients diagnosed with low-risk papillary thyroid cancer (pT1-pT2, radically resected at least 6 months prior to sampling, with no distant metastases), who required no adjuvant radioiodine therapy, with no other malignancies at the time of inclusion. Controls were euthyroid on the date of sampling, defined by thyroid-stimulating hormone (TSH) levels measured between 0.5 and 2 mIU/L. The controls were sampled on admission and all the other exclusion criteria from the test group were also applied to the controls.

### 2.2. Laboratory Analysis

Venous blood samples were collected from NET patients during routine follow-up visits (prior to lanreotide administration) and from controls on hospital admission. Samples were pseudonymized and technicians performing the assays were blinded to clinical data. The samples were processed within 6 h of collection, centrifuged, and stored at −80 °C until analysis.

Serum visfatin concentrations were measured using the Human Visfatin ELISA Kit (Invitrogen by ThermoFisher Scientific, Waltham, MA, USA) with a detection range of 1.1–300 ng/mL. Each sample was analyzed twice, and the mean value was calculated. Values outside the assay range were recorded as 1.1 ng/mL (minimum) or 300 ng/mL (maximum).

### 2.3. Statistical Calculations

Normality was assessed via the Shapiro–Wilk test. Group comparisons were performed using Mann–Whitney U (two groups) or Kruskal–Wallis (multiple groups) tests. Associations between visfatin and age were evaluated via Spearman’s rank correlation. Diagnostic performance was assessed using ROC curves, with optimal thresholds determined by Youden’s J statistic. Multiple linear regression tested associations between visfatin and clinical variables (age, sex, primary site, grade). Due to non-normally distributed data, continuous variables are reported as the median [IQR]. All tests were two-tailed with *p* values below 0.05 considered statistically significant. Statistical calculations were performed in Python 3.12 using the matplotlib 3.10.0, openpyxl 3.2.0 b1, pandas 2.2.3, patsy 1.0.1., pingouin 0.5.5, scikit-learn 1.6.1, scikit-posthocs 0.11.2, scipy 1.15.1, seaborn 0.13.2, and statsmodels 0.14.4 packages.

## 3. Results

### 3.1. Baseline Characteristics

We recruited 77 patients with non-resectable NETs, including 33 with panNETs and 44 with siNETs, and 29 controls. Both panNET and siNET patients presented a similar sex and age distribution, with the controls being younger than the patients with NETs and being predominantly female. The baseline study and control group demographics and clinical characteristics are summarized in Table 1.

### 3.2. Serum Visfatin Concentrations in NETs vs. Controls

Median [IQR] serum visfatin levels were higher in patients with NETs regardless of tumor’s primary site (6.94 [2.11–236.17] ng/mL, *p* = 0.004) as well as when stratified by panNETs (4.98 [2.13–264.96] ng/mL, *p* = 0.019) and siNETs (7.46 [2.01–199.44] ng/mL, *p* = 0.007) vs. controls (1.59 [1.1–9.24] ng/mL). In NET patients, visfatin concentrations varied (IQR: 234.06 ng/mL), with 16 (20.8%) patients above and 14 (18.2%) below the assay’s reference range (1.1–300 ng/mL). In controls, only 1 (3.4%) subject exceeded the reference range, while 13 (44.8%) were below it. Detailed results of the analysis are presented in Table 2 and Figure 1 and the complete statistical analysis can be found in the Appendix A.

### 3.3. Subgroup Comparison and Correlation Analysis

Subgroup analyses were performed to determine if visfatin levels varied by NET primary site, WHO grade, or sex. We found no differences in visfatin concentrations depending on NET primary site (panNETs vs. siNETs, median [IQR]: 4.98 [2.13–264.96] vs. 7.46 [2.01–199.44] ng/mL, *p* = 0.95), tumor grade (G1 vs. G2, median [IQR]: 17.23 [2.44–265.55] vs. 4.45 [1.62–158.45] ng/mL, *p* = 0.31), and sex (male vs. female, median [IQR]: 20.27 [1.23–277.34] vs. 4.84 [2.42–104.56] ng/mL, *p* = 0.89). Furthermore, when stratified by both primary site and grade, the Kruskal–Wallis test revealed no significant variation in visfatin levels (*p* = 0.18). Serum visfatin did not correlate significantly with age in NET patients (R Spearman = −0.17, *p* = 0.13). Finally, multiple linear regression (adjusting for age, sex, primary site, and grade) confirmed no impact of these variables on serum visfatin (R^2^ = 0.036; all *p* > 0.2). The results are presented in Table 3 and Table 4 and Figure 2 and Figure 3.

### 3.4. Serum Visfatin’s Diagnostic Performance

The diagnostic utility of serum visfatin for distinguishing NET patients from controls was evaluated using ROC curve analysis. The analysis revealed a moderate area under the curve (AUC) value of 0.68 (0.59–0.77). A cutoff value of 2.11 ng/mL was established with Youden’s index, providing a sensitivity of 75.3% (65.7–85.0%) and a specificity of 58.6% (40.7–76.6%). At this cut-off, the positive predictive value (PPV) was 82.86% (74.0–91.7%) and the negative predictive value (NPV) was 47.22% (30.9–63.5%). The ROC curves are displayed in Figure 4.

The complete results of the statistical analysis are included in Appendix A.

## 4. Discussion

Our study reveals that serum visfatin levels are increased in patients with NETs compared to controls without an active malignancy. ROC curve analyses revealed moderate diagnostic performance for visfatin across all patients with NETs and when divided into panNETs and siNETs. Notably, visfatin levels were consistent in NET subgroups determined by patient and tumor characteristics. Given the absence of prior studies of serum visfatin in NETs, these results have to be discussed in the broader context of research in other malignancies and in tumor tissue.

### 4.1. Overview of Visfatin and Its Role in Tumorigenesis

eNAMPT/visfatin functions primarily as a proinflammatory cytokine/adipocytokine. It exerts its responses via membrane receptors, though a precise mechanism of action remains under investigation [7]. Visfatin also participates in glucose homeostasis, angiogenesis, and may function as an ectoenzyme. It is actively secreted via a non-classical pathway, stimulated by cellular stress (e.g., hypoxia and ischemia) as well as by inflammatory processes [7]. Various factors are suggested to increase circulating visfatin, including caloric intake, exercise, and circadian cycles. Table 5 displays examples of visfatin alterations in several pathological conditions [12,13]. Visfatin is involved in the adipose–malignancy interactions alongside other adipocytokines, such as resistin and leptin, and may contribute to increased neoplasm risk in obesity [14]. Emerging evidence points to a causative role of visfatin in breast cancer, liver cancer, and multiple myeloma [15,16,17]. Both eNAMPT and iNAMPT are generally considered oncogenic, although their role in cancer is complex. Elevated circulating visfatin may result from tumor cell secretion or overproduction within the tumor microenvironment [7,9]. NAMPT facilitates cancer cell proliferation via NAD+ supply, the activation of multiple signaling pathways, and apoptosis inhibition [7,8]. Additionally, visfatin promotes cancer cell migration and metastasis by upregulating SDF-1/CXCL12, TGF-β, and the gelatinases MMP-2/-9 [7].

### 4.2. Diagnostic Utility of Serum Visfatin in NETs

Our study revealed that patients with NETs exhibited significantly higher serum visfatin levels compared to the control group, regardless of the tumor’s primary site (Figure 1). ROC curve analysis showed moderate diagnostic performance of serum visfatin across all NET patients (AUC = 0.68; 95%CI: 0.59–0.77). A cut-off value of 2.11 ng/mL achieved a sensitivity of 75.3%, and a specificity of 58.6%. Subgroup analysis for panNET and siNET patients yielded similar ROC results, as shown in Figure 4.

Although circulating visfatin has not been previously researched in NETs, our results align with similar studies across various neoplasms. Increased circulating visfatin levels and NAMPT tissue overexpression was reported in several malignancies, although discrepancies exist between studies and between tissue and serum compartments (e.g., in thyroid cancer), as seen in Table 6 [10,40]. In NET animal models, visfatin is expressed in pituitary and gastrointestinal neuroendocrine cells and has a regulatory role in cell proliferation and hormone secretion [41,42,43]. NAMPT overexpression has also been observed in patient-derived panNET tissue, suggesting its potential role in tumor pathogenesis [44].

### 4.3. Association of Visfatin with Tumor and Patient Characteristics

Given the heterogeneity of NETs, we analyzed visfatin levels across NET primary sites, WHO Grades (panNETs vs. siNETs, G1 vs. G2), sex, and age. Although some trends were observed, there was no significant difference between variables, either individually or in composite groups (Table 3 and Table 4, Figure 2 and Figure 3). This consistency could be favorable in a biomarker setting. Contrasting with chromogranin A’s variability, serum visfatin may present a more consistent option, independent of tumor features [2].

In other malignancies, elevated visfatin levels were associated with advanced tumor size, stage, and disease progression, the presence of lymph node metastases, higher metastatic rates, and resistance to treatment [8]. However, most studies did not establish a significant correlation between visfatin concentrations and tumor histological grade [61]. A meta-analysis highlighted a consistent correlation between elevated serum visfatin and poor overall survival in endometrial, colorectal, breast, and bladder cancers [11,51,62]. Importantly, due to the cross-sectional design of this study, we were unable to assess the prognostic value of visfatin in NETs. Prospective studies are needed to explore these associations and clarify visfatin’s prognostic potential.

In previous studies, both positive and negative correlations between age/sex and serum visfatin can be found, with no significant sex differences [63,64]. Our study revealed no statistically significant differences between males and females and did not observe a correlation with age. These findings align with the perspective that underlying health conditions may have more impact on visfatin levels than sex or age.

### 4.4. Visfatin as a Therapeutic Target—NAMPT Inhibitors

Further evidence of NAMPT’s significance in neuroendocrine malignancies comes from studies on small cell lung cancer (SCLC), revealing a strong reliance of cancer cells on NAD+ synthesis through the NAMPT-dependent salvage pathway. SCLC and other cells of neuroendocrine differentiation could be particularly vulnerable to NAMPT inhibitors (NAMPT-i) [65]. This is supported by Audrito et al., suggesting that NAMPT overexpression correlates with tumorigenesis and may be utilized in NAMPT-i therapy [66]. Further preclinical studies have shown that NAMPT-i exhibit strong anti-tumor activity against neuroblastoma and SCLC cell lines. This effect is attributed to the high dependence of these tumors on oxidative phosphorylation for energy production and reduced expression of Yes-Associated Protein 1 (YAP1) [67]. One in vitro study suggests NAMPT-i as potential drug candidates for NET treatment [68].

Emerging data suggest that NAMPT-i could also improve existing NET therapies. PanNET cells frequently exhibit overactivity of the PI3K/AKT/mTOR pathway, and NAMPT has been linked with a resistance mechanism against everolimus, an mTOR inhibitor. Dual inhibition of NAMPT and mTOR may have a synergistic effect and enhance treatment efficacy [69]. Additionally, NAMPT inhibition sensitizes NET cells to radiation and has been shown to improve response to radioligand therapy with 177Lu-DOTATATE [70].

These findings highlight the potential of targeting NAMPT as a novel therapeutic option for NETs. Although circulating visfatin is not a direct substitute for NAMPT, its study can provide valuable insights into disease status and aid in personalizing treatment. While NAMPT-i are still in the early stages of development, some have already been researched in NENs and others have entered clinical trials, paving the way for their potential therapeutic implementation in the near future [70,71,72].

### 4.5. Limitations and Future Directions

This study has several limitations. As the first study examining serum visfatin levels in NETs, there is no previous data for direct comparison, and validation in an independent cohort is needed. NETs are heterogeneous and further analyses should include patients with additional primary tumor sites (e.g., pulmonary carcinoids or hindgut NETs). Moreover, our study was limited to G1 and G2 NETs. Including G3 NETs and comparing NETs with NECs would be valuable, as in vitro and pharmacological studies suggest that higher grade tumors may exhibit even higher visfatin levels [65,67].

Visfatin is affected by other comorbidities (Table 5), complicating the selection of an optimal control group. We minimized confounding factors by excluding subjects with conditions known to affect visfatin (e.g., diabetes, renal or liver failure, untreated hyperlipidemia, hypertension, thyroid dysfunction, recent surgery, or active malignancies other than NETs). Nevertheless, some selection bias cannot be entirely ruled out. Our control group consisted of patients with radically resected low-risk papillary thyroid cancer in a euthyroid state and healthy otherwise, without significant comorbidities. This selection appears justified, as previous research indicates that thyroid cancer is not associated with altered visfatin levels (indeed, a control group without any malignancy might exhibit even lower levels of visfatin) [10]. Moreover, the sex and age distributions of the controls differ from those of the test group. Although these factors do not appear to affect serum visfatin both within sex- and age-matched NET subgroups and in previous research, we acknowledge this as a limitation.

Finally, a cross-sectional study design prevents the evaluation of visfatin’s prognostic value in NETs. All NET patients were treated with SSA, the impact of which on serum visfatin remains unknown. In order to assess visfatin’s relationship with disease outcomes, treatment response, and tumor burden and to determine any treatment-related effects, future studies should include longitudinal measurements or compare treatment-naive patients with those undergoing therapy.

## 5. Conclusions

For the first time, we showed that serum visfatin levels have moderate diagnostic ability in distinguishing NETs from controls [73]. Visfatin is a monoanalyte and, as such, cannot describe disease status as comprehensively as next-generation biomarkers. However, it appears less variable across NET subgroups than other single biomarkers, such as chromogranin A. Visfatin/NAMPT plays an important role in cancer biology—in cell migration, tissue proliferation, and metabolic regulation—making it an attractive candidate both in biomarker research and as a therapeutic target. This study is limited by its single-center, cross-sectional design, which needs to be addressed in future studies through longitudinal measurements and larger patient cohorts. Nevertheless, visfatin’s promising results, combined with a low-cost approach, warrant further investigation.

## Figures and Tables

**Figure 1 cancers-17-02343-f001:**
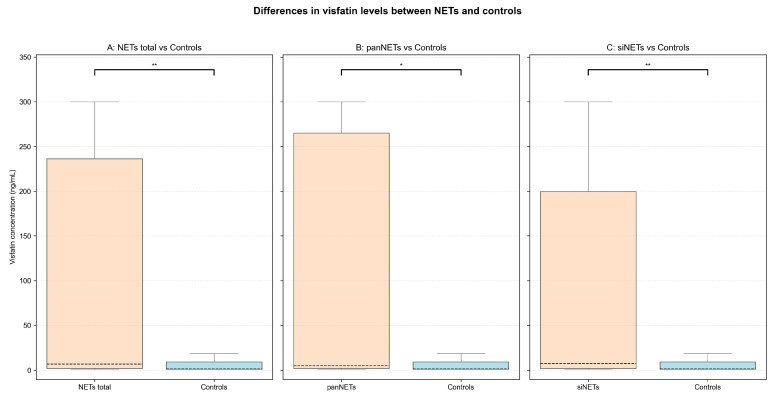
Differences in visfatin levels between NET patients and controls. The box and whiskers plots display the distribution of visfatin concentrations between (**A**) all patients with NETs (NETs total), (**B**) patients with pancreatic NETs (panNETs), (**C**) patients with small intestinal NETs (siNETs) and controls. The dashed line within the box indicates the median visfatin concentration in the group, with IQR highlighted by the edges of the box. Whiskers extend to 1.5 IQR. The significance level of each comparison is highlighted above the box plots: * *p* < 0.05, ** *p* < 0.01.

**Figure 2 cancers-17-02343-f002:**
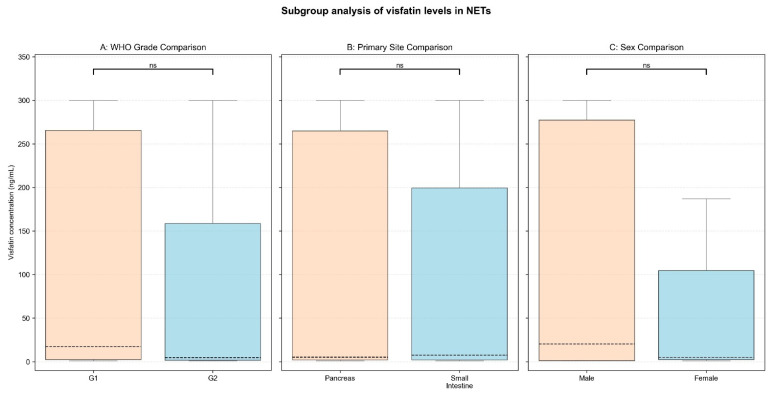
Differences in visfatin levels between NET subgroups. The box and whiskers plots display the distribution of visfatin concentrations determined by (**A**) WHO grade, (**B**) NET primary site, (**C**) NET patients’ sex. The dashed line within the box marks the median visfatin concentration in the group, with IQR highlighted by the edges of the box. Whiskers extend to 1.5 IQR. The significance level of each comparison is highlighted above the box plots: ns, non-significant.

**Figure 3 cancers-17-02343-f003:**
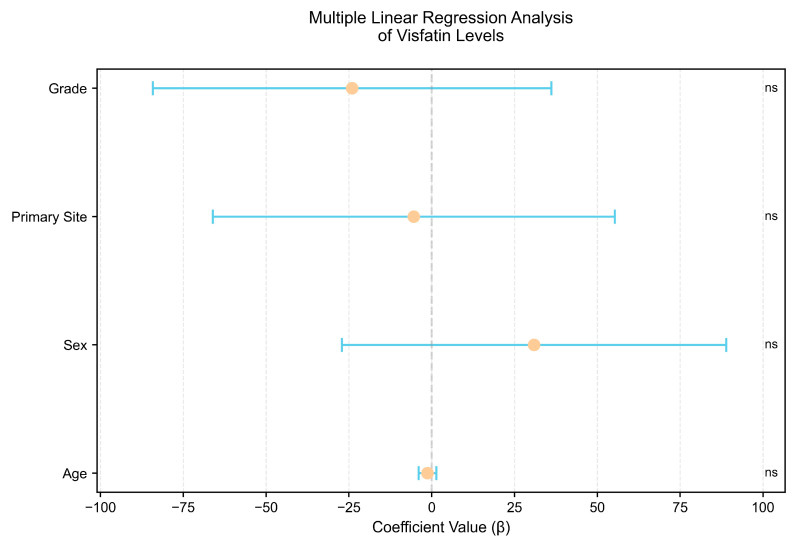
Multiple linear regression analysis of visfatin concentration in NET patients. The forest plot visualizes β-coefficients with 95% CI for each variable. The markers represent the regression coefficient, with whiskers marking the CI. The significance level of each comparison is highlighted to the right of each comparison: ns, non-significant.

**Figure 4 cancers-17-02343-f004:**
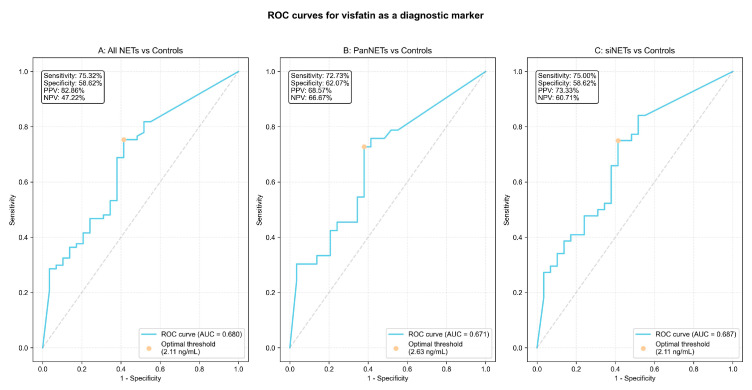
Diagnostic performance of serum visfatin in NET patients. The receiver operating characteristic curves display the performance of serum visfatin concentrations as a diagnostic biomarker in (**A**) all study NET patients, (**B**) pancreatic NET patients (panNETs), and (**C**) small intestinal NET patients (siNETs) vs. controls. The dot indicates the optimal threshold for sensitivity and specificity.

**Table 1 cancers-17-02343-t001:** Baseline demographics and clinical characteristics. The results are displayed as frequencies with percentages in parentheses, or as * the median [IQR].

Variable				NETs Patients			
				Total NETs (*n* = 77)	Pancreatic NETs (*n* = 33)	Small Intestinal NETs (*n* = 44)	Controls (*n* = 29)
**Demographics**							
	**Age, years**			71.0 [63.0–77.0] *	70.0 [59.0–78.0] *	71.0 [65.8–76.0] *	56.0 [43.0–64.0] *
	**Sex**						
		**Male**		39 (50.7%)	17 (51.5%)	22 (50%)	4 (13.8%)
		**Female**		38 (49.3%)	16 (48.5%)	22 (50%)	25 (86.2%)
**Clinical Characteristics**							
	**NET primary site**						
		**Pancreas**		33 (42.9%)			
		**Small Intestine**		44 (57.1%)			
	**Functioning NETs**						
		**Total**		26 (33.8%)	5 (15.2%)	21 (47.7%)	
			**Carcinoid Syndrome**	21 (27.3%)	*-*	21 (47.7%)	
			**Insulinoma**	4 (5.2%)	4 (12.1%)	-	
			**Glucagonoma**	1 (0.1%)	1 (0.3%)	-	
	**WHO Grade**						
		**G1**		35 (45.5%)	10 (30.3%)	25 (56.8%)	
		**G2**		42 (54.5%)	23 (69.7%)	19 (43.2%)	

**Table 2 cancers-17-02343-t002:** Serum visfatin levels in NET patients vs. controls.

Subgroup	Median [IQR] (ng/mL)		Mann–Whitney U Statistic	Effect Size (r)	*p*-Value
	NETs	Controls			
**Total NETs**	6.94 [2.11–236.17]	1.59 [1.10–9.24]	1518.5	0.36	0.004
**panNETs**	4.98 [2.13–264.96]		642.0	0.34	0.019
**siNETs**	7.46 [2.01–199.44]		876.5	0.37	0.007

**Table 3 cancers-17-02343-t003:** Subgroup comparisons and correlation analysis of serum visfatin in NET patients.

Variable	Subgroup	Median [IQR] (ng/mL)	Mann–Whitney U Statistic	Effect Size (r)	*p*-Value
**Primary Site**	panNETs	4.98 [2.13–264.96]	732.5	0.01	0.95
	siNETs	7.46 [2.01–199.44]			
**WHO Grade**	G1	17.23 [2.44–265.55]	834.0	0.11	0.31
	G2	4.45 [1.62–158.45]			
**Sex (NETs)**	Male	20.27 [1.23–277.34]	755.5	0.02	0.89
	Female	4.84 [2.42–104.56]			
	**Kruskal–Wallis H statistic**			**Degrees of freedom**	
**Primary Site * WHO Grade**	4.9124			3	0.18
	**R Spearman**				
**Age (NETs)**	−0.1731				0.13

**Table 4 cancers-17-02343-t004:** Multiple linear regression analysis of factors associated with serum visfatin levels in NET patients.

Variable	Regression Coefficient (5–95% CI)	Standard Error	*p*-Value
**Age**	−1.28 (−3.96–1.41)	1.35	0.35
**Sex (Male)**	30.88 (−27.14–88.90)	29.11	0.29
**Primary Site (siNET)**	−5.41 (−66.10–55.28)	30.44	0.86
**Grade (G2)**	−24.04 (−84.18–36.10)	30.17	0.43
	**R-squared**	**Adjusted R-squared**	
**Model Fit**	0.036	−0.018	

**Table 5 cancers-17-02343-t005:** Selected pathological conditions associated with elevated serum visfatin.

Conditions with Elevated Serum Visfatin		Study
**Inflammatory diseases**	Rheumatoid arthritis	Cheleschi [18] Ali [19]
	Osteoarthritis	Askari [20] Fioravanti [21]
	Inflammatory bowel disease	Colombo [22] Neubauer [23]
	Lung injury	Bime [24] Lee [25]
**Metabolic disorders**	Type 2 diabetes	Mir [26] Mostafa [27]
	Insulin resistance	Nourbakhsh [28] Chen [29]
	Obesity	Yin [30] Nourbakhsh [28]
	Metabolic syndrome	Zhong [31]
**Cardiovascular** **diseases**	Hypertension	Gunes [32] Liakos [33]
	Cerebrovascular accidents	Gu [34] Huang [35] Wang [36]
	Acute coronary syndrome	Zhang [37]
	Atherosclerosis	Zheng [38] Kadoglou [39]

**Table 6 cancers-17-02343-t006:** Visfatin/NAMPT alterations in blood and tumor tissue across various malignancies.

Study	Malignancy	Assessment	Alteration
Olesen [45]	Hematopoietic malignancies	Tumor	↑
Wang [46]	Endometrial cancer	Circulating	↑
Tian [47]	Endometrial cancer	Tumor	↑
Dalamaga [48]	Breast cancer	Circulating	↑
Folgueira [49] Kim [50]	Breast cancer	Tumor	↑
Zhang [51]	Bladder cancer	Circulating	↑
Sawicka-Gutaj [10]	Thyroid cancer	Circulating	↔
Sawicka- Gutaj [40]	Thyroid cancer	Tumor	↑
El-Daly [52]	Hepatocellular carcinoma	Circulating	↑
Fazeli [53] Kosova [54]	Colorectal cancer	Circulating	↑/↔
Hufton [55] Beijnum [56]	Colorectal cancer	Tumor	↑
Wang [57] Patel [58]	Prostate cancer	Tumor	↑
Bi [59] Long [60]	Gastric cancer	Tumor	↑

## Data Availability

Data is available as a Appendix A from the corresponding authors upon reasonable request.

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
