# Peer review of "Serum Visfatin/eNAMPT as a Biomarker in Pancreatic and Small Intestine Neuroendocrine Tumors: A Cross-Sectional Study and Future Perspectives"

_cancers, 2025, doi:10.3390/cancers17142343_

Round 1

Reviewer 1 Report

Comments and Suggestions for Authors

Komarnicki et al:

The authors present a study where they evaluate serum Visfatin in patients experiencing neuro-endocrine tumors. They find elevated levels of Visfatin in patients, suggesting that this compound could be used as a biomarker for NET's. They find no correlation between Visfatin levels and age or gender suggesting that Visfatin could be universal biomarker for such type of cancers. I think they make some sound arguments, and I have a few questions:

  1. Is Visfatin found to be elevated in other cancers? If so, could the authors claim that it is a specific biomarker for NET's? Or is it more of a biomarker for all cancers?
  2. Patients in their cohort were treated with long-acting somatostatin analogs. How does this treatment affect Visfatin levels? Is there a way to ascertain Visfatin levels without such treatment?
  3. In Fig 2C there seems to be a pretty visible difference between male vs. females. I understand that the authors found this difference to be not significant, but did they try multiple methods of statistical comparisons to reach that conclusion? An opinion of a statistician may be warranted here.

Author Response

Dear Reviewer,

thank you for your suggestions and for taking your time to review the manuscript. You can find our point-by-point replies below:

Comment 1: Is Visfatin found to be elevated in other cancers? If so, could the authors claim that it is a specific biomarker for NET's? Or is it more of a biomarker for all cancers?

Response 1: Visfatin was found to be elevated in other cancers – examples can be found in Table 6 in the Discussion section. Thus, it might be difficult for visfatin to be considered a NET-specific biomarker. Even then, visfatin could be a part of a diagnostic protocol, along with other monoanalytes to improve their standalone accuracy (such as e.g. chromogranin A) or used in a specific scenario (such as neuron-specific enolase in high grade NETs). Neither chromogranin A nor neuron-specific enolase are specific enough for screening, but have their own place in specific scenarios – so can visfatin, but that remains to be explored. This is also the main take away from the study, which we highlighted in the revised version of the article.

Comment 2: Patients in their cohort were treated with long-acting somatostatin analogs. How does this treatment affect Visfatin levels? Is there a way to ascertain Visfatin levels without such treatment?

Response 2: SSA impact on visfatin levels is yet to be explored, and remains the limitation of our study (as listed in final paragraph of section 4.5.). To evaluate visfatin levels without SSA treatment, measurements would have to be performed before first SSA dose. Considering NETs rarity, changing the study design to accommodate a similar test group structure would make patient recruitment more difficult and time consuming (between potentially a few years in a single NET specialized center, to approximately a year if performing a multi-center trial). Although not impossible, this would be more reasonable as a follow-up rather than an exploratory study (now that we have found the difference and acknowledged the limitations). Moreover, a small decrease or no impact of SSA has been previously observed in other monoanalytes such as chromogranin A or 5-HIAA. In visfatin, both of these outcomes would have little impact on the main takeaway of our study (the elevation of serum visfatin in NETs), but remain a limitation – as stated in the manuscript.

Comment 3: In Fig 2C there seems to be a pretty visible difference between male vs. females. I understand that the authors found this difference to be not significant, but did they try multiple methods of statistical comparisons to reach that conclusion? An opinion of a statistician may be warranted here.

Response 3: Thank you for this suggestion. Indeed, the difference may appear visible on the graph and we agree that statistical significance is not the only factor that we need to consider. However, several points of view are already presented in the main file and in the supplementary. Figure 2 and Table 3 present the results of Mann-Whitney U test (the difference in serum visfatin levels between NETs subgroups in non-normally distributed data). Figure 3 and Table 4 present this relationship from a perspective of multiple linear regression (the relationship between specific variables and visfatin levels). Both of these methods suggest that sex is not a relevant factor when measuring serum visfatin. Meanwhile, in the supplementary file, we have also included multiple linear regression of log-transformed visfatin and Mann-Whitney U test for visfatin and sex in control group - both confirm findings from the main file. Therefore, sex/visfatin relationship is consistent within our study and is also in line with previous literature findings (citations 63-64).

Reviewer 2 Report

Comments and Suggestions for Authors

In this study, the authors present, for the first time, the potential utility of serum visfatin/eNAMPT measurements as a diagnostic biomarker for Neuroendocrine Tumors (NETs). Given that NETs continue to pose significant challenges in the field of endocrine oncology, this research is particularly noteworthy. 

I would like to offer a few suggestions for consideration:

  1. It is advisable for the authors to remove the term "review" from the title, as the content of the article does not align with this designation, which may lead to confusion among readers.

  1. To maintain consistency with the data presented in the tables, it is recommended that the authors include the exact P values in Figures 1, 2, and 3.

  1. In Table 1, the authors should consider adding more detailed information regarding clinical characteristics, including treatment strategies, response rates, and survival outcomes.

  1. Based on my understanding, the sensitivity and specificity values presented in Figure 4 do not provide robust support for the conclusion that serum visfatin is a reliable diagnostic marker. I would appreciate an explanation from the authors regarding this matter, and I suggest that this point be addressed in the discussion section.

Author Response

Dear Reviewer,

Thank you for taking your time to evaluate our article and for your comments. I submit our replies below:

Comment 1: It is advisable for the authors to remove the term "review" from the title, as the content of the article does not align with this designation, which may lead to confusion among readers.

Response 1: We agree with the suggestion and revised the title.

Comment 2: To maintain consistency with the data presented in the tables, it is recommended that the authors include the exact P values in Figures 1, 2, and 3.

Response 2: The reason for the omission of exact p values was that the exact numbers are already presented in the tables that are adjacent to the respective figures just above them in the manuscript. Instead, to achieve a clear and simple data presentation, we deliberately used the asterisks/”ns” and explained their meaning in each figure’s description field. We have discussed this internally before revising and decided to retain the original data presentation, in order not to duplicate the same numbers in figures, tables, main text, abstract, and the supplementary file.

Comment 3: In Table 1, the authors should consider adding more detailed information regarding clinical characteristics, including treatment strategies, response rates, and survival outcomes.

Response 3: Thank you, we have added data on functioning NETs in both panNETs and siNETs to include more detail in the table. Regarding treatment strategies, all NETs patients were treated with Somatostatin Analogs, and other treatment methods (PRRT, alkylating chemotherapy, everolimus, sunitinib) were excluded. Consequently, we omitted them from the table and only placed them in the main text (section 2.1.), to avoid listing 0%/100% variables. Response rates and survival rates cannot be reported for the same reason, as all patients had SD at the time of the inclusion. Otherwise, if the patient presented with PD per RECIST 1.1, they would have been qualified for another line of treatment and excluded from this study. We consider this a reasonable approach to patient inclusion, as this is the first study of serum visfatin in NETs and we do not know how treatment affects its concentrations. Including multiple treatment modalities would dilute the study group and decrease the statistical power. Reporting patient survival after a set follow-up period would be useful in a prognostic/predictive biomarker study. However, this was not the aim of this research and we suggested it as a limitation and future research direction in section 4.5.

Comment 4: Based on my understanding, the sensitivity and specificity values presented in Figure 4 do not provide robust support for the conclusion that serum visfatin is a reliable diagnostic marker. I would appreciate an explanation from the authors regarding this matter, and I suggest that this point be addressed in the discussion section.

Response 4: We used the term “reliable” in section 4.3. to confront visfatin with high variability of chromogranin A within NETs subgroups. We agree that the values from Figure 4 are not necessarily reliable – therefore we used terms like “moderate ability” when talking about diagnostic accuracy and listed its inferiority to multianalyte assays in the discussion and the conclusions. We have modified the description of the results in sections 4.2. and 4.3. and corrected the wording to more accurately reflect our results.

Reviewer 3 Report

Comments and Suggestions for Authors

In this paper the authors assess the serum Visfatin concentration in patients with pancreatic and small intestine NETs, with the aim to correlate with patient and tumor characteristics.

Introduction: provide suficient information about the topic.

Materials and Methods: 

I consider that the number of patients with pancreatic and small intestine neoplasia is adequate.

The number of male cases in the control group is much lower than in the siNET and panNET groups. This may be an important factor in the analysis.

In controls the number of male and female is not balanced, and the age in control group is not comparable with siNETs and panNETs

Why are patients with low-risk papillary thyroid cancer included as a control group? It would be better to include normal or healthy patients, or patients with other malignancies unrelated to endocrine tumors.

Discussion

On line 311 the authors state that thyroid cancer is not associated with altered Visfatin levels, however there is a report by Tubart-Herrera Turbat-Herrera EA, Kilpatrick MJ, Chen J, Meram AT, Cotelingam J, Ghali G, Kevil CG, Coppola D, Shackelford RE. Cystatione β-Synthase Is Increased in Thyroid Malignancies. Anticancer Res. 2018 Nov;38(11):6085-6090. doi: 10.21873/anticanres.12958. PMID: 30396922; PMCID: PMC7771238.) in which they indicated that NAMPT immunohistochemistry may be useful in differentiating thyroid follicular adenomas from follicular carcinomas.

I suggest that the authors justify this observation, because the results were obtained using different tests.

Author Response

Dear Reviewer, thank you for taking your time to assess our manuscript and provide us with comments and suggestions. We have evaluated your proposals and reply to them below.

Comment 1: Introduction: provide suficient information about the topic.

Response 1: Thank you for the comment. We believe that the current introduction provides a concise and sufficient overview of this topic.  The section includes basic information about NETs in general, current consensus on optimal biomarkers (and their limitations in certain areas), basic information about visfatin in general and specifically in neoplasms, our aims and reasoning behind this study. A more detailed explanation can be found in the discussion, as we wanted to keep the manuscript organized and streamlined. However, if there is anything specific that you consider missing, we would be happy to address it.

Comment 2: Materials and Methods:

I consider that the number of patients with pancreatic and small intestine neoplasia is adequate.

The number of male cases in the control group is much lower than in the siNET and panNET groups. This may be an important factor in the analysis.

In controls the number of male and female is not balanced, and the age in control group is not comparable with siNETs and panNETs

Why are patients with low-risk papillary thyroid cancer included as a control group? It would be better to include normal or healthy patients, or patients with other malignancies unrelated to endocrine tumors.

Response 2: Thank you for this detailed analysis. We agree that even though sample size is sufficient, uneven sex and age distribution has to be considered a limitation of the study. We have addressed this in a limitations section and included in the revision. However, neither of these variables appear to impact serum visfatin in NETs subgroups analyses (which are sex- and age- matched) and in prior research. Regarding the choice of controls, we have deliberately chosen low-risk PTC, due to their availability at our department and complex diagnostic process on diagnosis (which regular controls would not undergo). This way, we were able to apply exclusion criteria more accurately, as we were able to screen for hyperlipidemia, thyroid dysfunction, hypertension, diabetes, and liver or kidney dysfunction, some of which might have been undiagnosed in regular controls – we have clarified it the revision. Most importantly, at the time of the sampling the controls had no active malignancy, as they underwent radical tumor resection, were non-metastatic, euthyroid, and had no indications for radioactive iodine therapy or any other therapeutic follow-up. Thyroid cancer in general is relatively common in the population (one meta-analysis reports 11% prevalence, some reports are even higher 10.1200/JCO.2016.67.7419).  Obviously, this is still not an ideal control group and we addressed it in the limitations. However, even if visfatin levels were increased in this control group, we would expect the difference between them and NETs to narrow. This was not the case and we believe that including this control group did not fundamentally alter the study outcome.

Comment 3: Discussion

On line 311 the authors state that thyroid cancer is not associated with altered Visfatin levels, however there is a report by Tubart-Herrera Turbat-Herrera EA, Kilpatrick MJ, Chen J, Meram AT, Cotelingam J, Ghali G, Kevil CG, Coppola D, Shackelford RE. Cystatione β-Synthase Is Increased in Thyroid Malignancies. Anticancer Res. 2018 Nov;38(11):6085-6090. doi: 10.21873/anticanres.12958. PMID: 30396922; PMCID: PMC7771238.) in which they indicated that NAMPT immunohistochemistry may be useful in differentiating thyroid follicular adenomas from follicular carcinomas.

I suggest that the authors justify this observation, because the results were obtained using different tests.

Response 3: Thank you for pointing us towards this study. This article focuses on NAMPT immunohistochemistry, which does not necessarily translate to serum concentrations. In thyroid cancer, there have been other reports about NAMPT tissue overexpression in thyroid cancer that we have listed (Table 6, reference no 40). However, when analyzing serum visfatin levels, no difference was found in thyroid cancer (this is also presented in the same table and discussed earlier in the manuscript: reference no 10). Considering that the suggested article is repeating the findings that are already referenced and discussed in the manuscript, we have chosen not to cite it, as the manuscript is already over the suggested reference threshold.

Thank you yet again for your advice and contributing to the improvement of the manuscript. We are looking forward to your reply and to address any further comments or suggestions that you might have. 

Round 2

Reviewer 3 Report

Comments and Suggestions for Authors

All doubts have been clarified.

However, it would have been advisable to include samples from patients with origins other than neuroendocrine tumors.

In the future, continuing on this topic, I recommend to evaluate serum visfatin levels and tissue expression using immunohistochemistry, including samples from normal patients as well as samples from other tumor types.